# Variations in the Frequencies of Polymorphisms in the CYP450s Genes in Eight Major Ethnicities of Iran: A Review of the Human Data

**DOI:** 10.3390/jpm12111848

**Published:** 2022-11-05

**Authors:** Navid Neyshaburinezhad, Hengameh Ghasim, Mohammadreza Rouini, Youssef Daali, Yalda H. Ardakani

**Affiliations:** 1Biopharmaceutics and Pharmacokinetic Division, Department of Pharmaceutics, Faculty of Pharmacy, Tehran University of Medical Sciences, Tehran 14176-14411, Iran; 2Division of Clinical Pharmacology and Toxicology, Geneva University Hospitals, 1205 Geneva, Switzerland; 3Faculty of Medicine, University of Geneva, 1206 Geneva, Switzerland

**Keywords:** Iranian ethnicities, CYP450 genotype, polymorphism, SNP

## Abstract

Genetic polymorphisms in cytochrome P450 genes can cause variation in metabolism. Thus, single nucleotide variants significantly impact drug pharmacokinetics, toxicity factors, and efficacy and safety of medicines. The distribution of CYP450 alleles varies drastically across ethnicities, with significant implications for personalized medicine and the healthcare system. We combined whole-genome and exome sequencing data to provide a review of CYP450 allele polymorphisms with clinical importance. Data were collected from 800 unrelated Iranians (100 subjects from 8 major ethnicities of Iran), more than 32,000 unrelated Europeans (other than Caucasian), and four Middle Eastern countries. We analyzed the frequencies and similarities of 17 CYP450 frequent alleles related to nine important CYP450 isoenzymes and homozygous and heterozygous genotypes based on these alleles in eight major Iranian ethnics by integrating these data with population-specific linkage information and compared these datasets with mentioned populations.

## 1. Introduction

Personalized medicine is one of the highly advanced fields of medical sciences. This branch of science delves into the patient’s physiologic and pathophysiologic features and selects and carries out an appropriate and specific therapeutic approach for the patient [1]. Concentrating on the patient as the focal point rather than the disease augments the treatment effectiveness and decreases the probable side effects [2]. Despite the positive aspects of this system, achieving such an extent of knowledge of the patient’s features to implement personalized medicine necessitates a tremendous amount of detailed information about patient physiology and pathophysiology [3,4]. The genetic assessment and the specific genes of each individual are two of the most significant characteristics that help us achieve this level of knowledge of the patient’s features [5]. So far, there have been numerous studies regarding the matter at hand. Primarily, an effort has been made to genetically assess the specific populations, such as the people of a country, and compare them to other countries [6,7,8]. Following that, the ethnicities of various countries have been studied and compared [9,10,11]. Such assessments/studies greatly help access the required information and individualized therapies. It also improves our understanding of the population’s genetic characteristics and the similarities and differences between individuals. The final stage of personalized medicine is thoroughly understanding each individual and identifying the physiologic and pathophysiologic similarities and disparities between patients [12]. It should be noted, however, that genetic assessment alone is insufficient to realize personalized medicine fully; phenotypic assessment is also crucial. Numerous studies have demonstrated that a patient’s response to a therapeutic approach or a prescribed pharmaceutical is not commensurate with his or her genetic profile [13,14,15].

Cytochrome p450 (CYP450) is one of the most potent metabolizing enzymes in the body. It metabolizes many endogenic and exogenic substances [16,17,18]. Differences in genotype and phenotype alter the enzyme’s activity. Aside from intra-individual changes in these enzymes’ activity by several physiological, pathological, and environmental factors [19,20,21,22,23,24,25], any change in the enzyme activity may alter pharmacotherapeutic efficiency and decrease or increase adverse drug reactions [26,27,28,29]. Hence, in personalized medicine, the enzyme family is one of the parameters that increase the specificity of pharmacotherapy in patients under investigation. A genetic assessment of CYP450 enzymes reveals that they fall into four categories: poor, intermediate, extensive, and ultra-rapid metabolizers, which have low, moderate, usual, and high activity, respectively [30]. As previously stated, the patient’s response to therapy is not always consistent with his genotype. For instance, the patient’s genotype indicates that the CYP450 enzyme is normal, but the pharmacotherapy response indicates the enzyme’s decreased activity [31]. In such cases, it has been demonstrated that an enzyme’s activity and phenotype are not concordant with the corresponding genotype for various reasons, such as diseases, medication use, diet, and the like. This phenomenon is known as phenocoversion. Nonetheless, the primary assessment of personalized medicine begins with genotyping [32,33].

In recent years, several studies have been conducted on the genotype of some cytochrome 450 enzymes in the Iranian population. In these studies, one cytochrome was investigated in one group of patients or a specific ethnicity [11,34,35,36]. However, no study has comprehensively investigated the important cytochromes P450 across various ethnicities in Iran’s population. In a review study, our research team studied the most critical enzymes of the cytochrome P450 family in the Iranian population and compared the results to those of the world’s five most significant populations (Europeans, Americans, Latinas, eastern and middle east Asia, and the Caucasian ethnicity). Results revealed that, while there are some similarities, there are many genotype differences between the cytochrome P450 enzyme of the Iranian population and the other populations mentioned above [37]. These findings highlighted the importance of conducting additional individual studies on the Iranian population to gain more information. Such data can advance personalized medicine, improve pharmacotherapy efficiency, and reduce probable unintended adverse effects in Iranian patients (concerning the genetic profile of cytochrome P450 enzymes).

For this reason, in this review article, our team investigated the genotype profile of the most critical cytochrome P450 enzymes in Iran across various ethnicities. Approximately 90 percent of drugs are metabolized through these enzymes. These enzymes include CYP1A2، CYP2B6، CYP2C8، CYP2C9، CYP2C19، CYP2D6، CYP2E1، CYP3A4, and CYP3A5. According to the study results, Iran’s population is divided into eight ethnicities of Arab, Azeri, Baloch, Kurd, Lur, Persian, Persian Gulf Islander, and Turkmen. Each group consisted of 100 healthy people who were studied. The required information was extracted, classified, and analyzed from the Iranome database, other available human genome variation databases, and a literature review. To our knowledge, the resulting data provide an immense overview of CYP450 allele distributions in Iranian ethnicities. They provide essential information for guiding ethnic-specific genotyping strategies in pharmacokinetic studies and, eventually, in personalized medicine.

## 2. Methods

### 2.1. Allele Frequency Data

The frequency of 17 alleles of eight cytochromes (CYP3A4 without SNP with a frequency above 0.5% among the Iranian population) was investigated in the current study using data from 8 main Iranian ethnicities (800 subjects) on the Iranome website. Furthermore, the frequency of these alleles among Iranian ethnicities was compared to the European population (32,299 Genomes) using the website [https://gnomad.broadinstitute.org, accessed on 10 August 2022]. It also was compared to the Caucasian race and four Middle Eastern countries’ (Egypt, Saudi Arabia, Turkey, and United Arab Emirates) populations through the related papers (50–250 people reported in each paper) (Table 1).

### 2.2. Allele Nomenclature and Definitions

CYP star (*) alleles were defined according to the Human CYP Allele Nomenclature Database (http://www.pharmvar.org, accessed on 10 August 2022) and (http://www.SNPedia.com, accessed on 10 August 2022). Table 2, Table 3 and Table 4 compare high-frequency alleles (SNPs) among major Iranian ethnicities and other populations (European, Middle Eastern) and Caucasian races. Table 5 and Table 6 show the frequency of CYP450s homozygous and heterozygous genotypes based on the most frequent SNPs in major ethnicities and the Iranian population (mean). Differences of less than 1% were considered similar. Lastly, the rate of homozygote and heterozygote genotypes of these alleles in Iranian ethnicities was compared.

In this regard, *common alleles* were defined as having minor allele frequency (MAF) > 1%. In contrast, rare variants or alleles had an allelic frequency ranging from 0.1% to 1%, and those with a frequency less than 0.1% were considered zero.

## 3. Results and Discussion

Several studies have reported the frequency of different CYP alleles in selected populations of Iranian healthy volunteers or patients [56,57,58]. However, most researchers have concentrated on a specific group of single nucleotide variants (SNVs) in each gene and studied the frequency of the alleles in a small group of mentioned populations [59,60,61,62]. In addition, relying on the findings of these studies can be problematic due to the disparities in genotyping methods and designed assay panels [11,63,64,65].

Recently, a comprehensive review was performed on 173 SNPs in the Iranian population. Using the Iranome website, https://gnomad.broadinstitute.org, accessed on 10 August 2022, and a literature review [37], the prevalence of the most frequent alleles was compared to the world’s five significant populations and the Caucasian race. Given all of the preceding, the Iranome database consisting of a whole exome sequencing of 800 individuals from Iran’s eight major ethnic groups (Arab, Azeri, Baloch, Kurd, Lur, Persian, Persian Gulf Islander, and Turkmen) was considered the primary source of data in this review. These data allow an accurate analysis of the genetic polymorphisms of the necessary CYP450 enzymes across Iranian ethnicities. In the next step, extracted data were compared to the Caucasian race and European and Middle Eastern populations using a literature review and other available databases.

In this regard, 17 SNPs associated with nine important CYP450 isoenzymes (CYP1A2, CYP2B6, CYP2C8, CYP2C9, CYP2C19, CYP2D6, CYP2E1, CYP3A4, and CYP3A5) were analyzed and will be discussed in greater detail.

### 3.1. CYP1A2

A previous study demonstrated that the frequency of the *1 allele in the Iranian population was 39.8% [37]. The only SNP with a frequency greater than 1% in the Iranian population is the *1F allele, which has a frequency of 59.9%. The Persian (66.8%) and the Persian Gulf Islander (50%), among the eight Iranian ethnicities, have the highest and lowest frequency, respectively (Table 2 and Figure 1). 

There are no similarities in the frequency of the *1F allele between Iranian ethnicities and any other studied populations (Caucasian, European, Middle Eastern) (difference more significant than 1%) (Table 3 and Table 4) [44,47,55]. Additionally, genotypic analysis of these ethnicities revealed that the Kurds (51.5%) and the Persian Gulf Islanders (31.6%) have the highest and lowest frequency of homozygote genotype of CYP1A2*1F, respectively. In comparison, the Baloch (42.4%) and Kurds (28.8%) had the highest and lowest heterozygote genotype frequencies, respectively (Table 5 and Figure 2). In the Iranian population, the mean frequency of CYP1A2 homozygous and heterozygous genotypes based on the *1F allele were 41.5% and 36.6%, respectively (Table 6 and Figure 2).

### 3.2. CYP2B6

The *1 allele was found in 54.7% of the Iranian population [37]. The *9 allele had the highest prevalence (26.6%) in the Iranian population, ranging from 30.5% in the Lur to 24.0% in the Persian (Table 2, Figure 1). The frequency of this allele among the Iranian Arabs was similar to the Caucasian race [66], the Persian similar to the European population (Table 3), and the Lur to the Middle Eastern population (Table 4) [40,45,52]. Additionally, genotypic investigation of these ethnicities revealed that the Lur (9%) and Persian (5%) had the highest and lowest frequency of homozygote genotypes, respectively. In comparison, the Lur (43%) and the Persian (38%), respectively, had the highest and lowest frequency of heterozygote genotypes (Table 5 and Figure 2). In the Iranian population, the mean frequency of CYP2B6 homozygous and heterozygous genotypes based on the *9 allele were 6.2% and 40.6% (Table 6 and Figure 2).

The *5 allele was the second most prevalent allele (8.5%) among the Iranian population, ranging from 10% in the Arab and Baloch to 6.5% in the Persian (Table 2 and Figure 1). This allele frequency among Arab and Baloch of the Iranian population was similar to the Caucasian race [66]. In contrast, none of the Iranian ethnicities were similar to the European population (Table 3) or the Middle Eastern population (Table 4) [40,45,52]. Furthermore, genotypic analysis of these ethnicities revealed that the Baloch (2%) had the highest frequency of homozygote genotype, while Turkmen, Lur, Far, and Persian Gulf Islander did not. On the other hand, among the Iranian population, Arab and Turkmen (18%) and Persian (13%) had the highest and lowest frequency of heterozygote genotype, respectively (Table 5 and Figure 2). The mean frequency of CYP2B6 homozygous and heterozygous genotypes based on the *5 allele were 0.6% and 15.7% in the Iranian population, respectively (Table 6 and Figure 2).

Among Iranian ethnics, the *2 allele prevalence ranged from 8.5% in the Persian to 1.5% in the Turkmen (Table 2 and Figure 1). The frequency of the *2 allele among Baloch and Lur of the Iranian populations was similar to the Caucasian race and the European populations (Table 3) [66]. Moreover, the Arabs were similar to the Middle Eastern population in terms of the *2 allele frequency (Table 4) [40,45,52]. Additionally, a genotypic investigation of Iranian ethnics revealed that Fars, Arab, Kurd, and Baloch had a 1% frequency of homozygote genotype, whereas other ethnicities did not have this genotype. On the other hand, among the Iranian population, the Persian (15%) and the Turkmen (3%) had the highest and lowest frequency of heterozygote genotype, respectively (Table 5 and Figure 2). Mean frequency of CYP2B6 homozygous and heterozygous genotype based on the *2 allele was 0.5% and 9.6% in the Iranian population (Table 6 and Figure 2).

The *22 allele had a 2.6% frequency in the Iranian population. It ranged from 4.5% among the Persians to 1% among the Persian Gulf Islanders (Table 2 and Figure 1). This allele’s frequency was similar to that of the Caucasian race, except for Persian and Persian Gulf Islanders [67]. Moreover, the frequency of this allele was similar to the European population of Kurd, Lur, and Persian Gulf Islanders (Table 3). Furthermore, a genotypic investigation of these ethnicities revealed that the homozygote genotype was not found among any of the Iranian ethnicities. The Persian (95) and the Persian Gulf Islander (2%), on the other hand, had the highest and lowest frequency of heterozygote genotype among the Iranian population, respectively (Table 5 and Figure 2). In the Iranian population, the mean frequency of CYP2B6 homozygous and heterozygous genotypes based on the *9 allele was 0 and 5.2%, respectively (Table 6 and Figure 2).

The *3 allele had the lowest prevalence (1.3%) in the Iranian population, ranging from 4% in the Persians to 0% in the Baloch (Table 2 and Figure 1). The frequency of this allele in the Azeri, Persian Gulf Islander, and Turkmen populations was similar to that of the Caucasian race and the European population (Table 3) [66]. Similar to the *22 allele, genotypic analysis of these ethnicities revealed that the homozygote genotype was not found among any Iranian ethnicity. On the other hand, the Persian race (8%) had the highest frequency of heterozygote genotype. In comparison, the Baloch showed no heterozygote genotype (Table 5 and Figure 2). In the Iranian population, the mean frequency of CYP2B6 homozygous and heterozygous genotypes based on the *3 allele was 0 and 2.6%, respectively (Table 6 and Figure 2).

### 3.3. CYP2C8

According to the previous study, the frequency of the *1 allele was 95% in the Iranian population [37]. The *4 allele prevalence ranged from 5.5% in the Kurds to 0.5% in the Persians among Iranian ethnicities (Table 2 and Figure 1). None of the Iranian ethnicities had a frequency of the *4 allele similar to the Caucasian race [68]. The frequency of this allele among the Kurds was similar to the European population (Table 3). Moreover, Arab, Lur, and Turkmen populations were similar to the Middle Eastern population regarding the *4 allele frequency (Table 4) [50]. Additionally, a genotypic study of Iranian ethnicities revealed that only the Azeri race had a 1% frequency of homozygote genotype, whereas the homozygote genotype was not found in other ethnicities. On the other hand, the Kurds (11%) and the Persian Gulf Islanders (1%) had the highest and lowest frequency of heterozygote genotype among the Iranian population, respectively (Table 5 and Figure 3). In the Iranian population, the mean frequency of CYP2C8 homozygous and heterozygous genotypes based on the *4 allele was 0.1% and 5.1%, respectively (Table 6 and Figure 3).

The prevalence of the *2 allele ranged from 6% in the Persian Gulf Islanders to 0% in Turkmen (Table 2 and Figure 1). The frequency of the *2 allele among the Lur was similar to the Caucasian race and the European population (Table 3) [54]. Additionally, a genotypic study of Iranian ethnics revealed that only Persian Gulf Islanders and Baloch had a 1% frequency of the homozygote genotype. Other ethnicities, on the other hand, did not have the homozygote genotype. In the Iranian population, the Persian Gulf Islanders (10%) had the highest frequency of heterozygote genotype. In comparison, the Turkmen lacked the heterozygote genotype (Table 5 and Figure 3). In the Iranian population, the mean frequency of CYP2C8 homozygous and heterozygous genotypes based on *2 allele was 0.2% and 4.2%, respectively (Table 6 and Figure 3).

### 3.4. CYP2C9

According to our previous study [37], the *1 allele had a 78.1% prevalence. Among Iranian ethnicities, the *2 allele prevalence ranged from 15% in Lur to 7% in Arab (Table 2 and Figure 1). None of the Iranian ethnicities had the same frequency of the *2 allele as the Caucasian race. However, the frequency of this allele among Azeris was similar to the European population (Table 3) [55]. Moreover, regarding *2 allele frequency, the Azeri and Turkmen were similar to the Middle Eastern population (Table 4) [40,41,43,46]. Additionally, a genotypic study of Iranian ethnics revealed that Persian and Persian Gulf Islanders had the highest homozygote genotype (2%), while Lur, Turkmen, Kur, and Baloch did not. However, among the Iranian population, the Lur (30%) and Arab (12%) had the highest and lowest frequency of heterozygote genotype, respectively (Table 5 and Figure 3). In the Iranian population, the mean frequency of CYP2C9 homozygous and heterozygous genotypes based on the *2 allele was 0.7% and 19.5% (Table 6 and Figure 3).

Among Iranian ethnicities, the *3 allele prevalence ranged from 14.5% in Baloch to 7% in Arabs (Table 2 and Figure 1). None of the Iranian ethnicities had a frequency of the *2 allele similar to the Caucasian race. However, the frequency of this allele among the Azeri and Arab populations was similar to the European population (Table 3) [55]. Moreover, except for Baloch and Persian Gulf Islanders, all Iranian ethnicities were similar to the Middle Eastern population in terms of the *3 allele frequency (Table 4) [40,41,43,46]. Additionally, a genotypic study of Iranian ethnicities revealed that Baloch had the highest frequency of homozygote genotype (5%), while Arab and Azeri did not. On the other hand, the Persian Gulf Islanders (21%) and Arab and Kurds (14%) had the highest and lowest frequency of heterozygote genotype among the Iranian population, respectively (Table 5 and Figure 3). In the Iranian population, the mean frequency of CYP2C9 homozygous and heterozygous genotypes based on the *3 allele was 1.7% and 17%, respectively (Table 6 and Figure 3).

### 3.5. CYP2C19

The frequency of the *1 allele in the Iranian population was 85.8% [37]. The only allele with a frequency of more than 1% among the Iranian population was the *2 allele. Its prevalence ranged from 18% in the Baloch to 11% in the Azeri, Lur, Kurd, and Persian Gulf Islanders (Table 2 and Figure 1). The frequency of the *2 allele was similar in the Arabs and Caucasians. However, among the Persian and Turkmen, its frequency was similar to the European population (Table 3) [55]. Moreover, the Baloch was similar to the Middle Eastern population regarding the *2 allele frequency (Table 4) [40,45,46,49]. Additionally, genotypic analysis of these ethnicities revealed that the Baloch (4%) and Fars, Arab, and Lur (2%) had the highest and lowest frequency of homozygote genotypes, respectively. In comparison, the Persian (29%) and Persian Gulf Islander (16%) populations had the highest and lowest frequency of heterozygote genotype, respectively (Table 5 and Figure 3). In the Iranian population, the mean frequency of CYP2C19 homozygous and heterozygous genotypes based on the *2 allele was 2% and 22.3%, respectively (Table 6 and Figure 3).

### 3.6. CYP2D6

A previous study found that the frequency of the *1 allele in the Iranian population was 10.6% [37]. The prevalence of the *2 allele was the highest in the Iranian population (47%), ranging from 53% in Baloch to 42% in Azeri (Table 2 and Figure 4). The frequency of the *2 allele among all Iranian ethnics was similar to the Caucasian race (Table 3) [55]. However, none were similar to the Middle Eastern population (Table 4) [39,48,51]. Furthermore, a genotypic analysis of these ethnicities revealed that the Baloch (32%) and Azeri (17%) had the highest and lowest frequency of homozygote genotype, respectively. In comparison, the Azeri (50%) and Lur (38.3%) had the highest and lowest frequency of heterozygote genotype, respectively (Table 5 and Figure 5). In the Iranian population, the mean frequency of CYP2D6 homozygous and heterozygous genotypes based on the *2 allele was 24.4% and 45.3%, respectively (Table 6 and Figure 5).

The *10 allele prevalence ranged from 22.5% in Azeris to 10% in the Arabs (Table 2 and Figure 4). None of the Iranian ethnics had the same frequency of the *10 allele as the Caucasian race, the European population (Table 3) [55] or the Middle Eastern population (Table 4) [39,48,51]. Additionally, genotypic analysis of these ethnicities revealed that the Azeri (7%) and Lur, Arab, and Persian Gulf Islander (1%) had the highest and lowest frequency of homozygote genotype, respectively. In comparison, the Azeri (31%) and Baloch (17%) had the highest and lowest frequency of heterozygote genotype, respectively (Table 5 and Figure 5). In the Iranian population, the mean frequency of CYP2D6 homozygous and heterozygous genotypes based on the *10 allele was 3.1% and 24%, respectively (Table 6 and Figure 5).

The *41 allele prevalence ranged from 18.5% in the Persian to 8.5% in the Persian Gulf Islander (Table 2 and Figure 4). The frequency of the *41 alleles among Turkmen and Persian Gulf Islander was similar to the Caucasian race and Turkmen to the European population (Table 3) [53]. Furthermore, except for Turkmen and Persian Gulf Islanders, the frequency of this allele was comparable to the Middle Eastern population (Table 4) [39,48,51]. Additionally, a genotypic analysis of these ethnicities revealed that Azeri, Kurd, Lur, and Baloch (4%) and Arab (1%) had the highest and lowest homozygote genotypes, respectively. In comparison, Azeri (29%) and Baloch (11%) had the highest and lowest frequency of heterozygote genotype, respectively (Table 5 and Figure 5). In the Iranian population, the mean frequency of CYP2D6 homozygous and heterozygous genotypes based on the *41 allele was 3.1% and 21.7%, respectively (Table 6 and Figure 5).

With a prevalence of more than 1%, the *4 alleles had the lowest frequency among CYP2D6 alleles (Table 2 and Figure 4). None of the Iranian ethnicities had the same frequency of the *4 allele as the Caucasian race or the European population (Table 3) [55]. However, the frequency of this allele among Kurds and Turkmen was similar to the Middle Eastern population (Table 4) [39,48,51]. Additionally, the Azeri (17%) and the Arab (7.5%) had this allele’s highest and lowest frequency. According to the genotypic analysis of these races, Azeri (4%) and Lur lacked the homozygote genotype. In comparison, Azeri (26%) and Turkmen (12.1%) had the highest and lowest frequency of heterozygote genotype, respectively (Table 5 and Figure 5). In the Iranian population, the mean frequency of CYP2D6 homozygous and heterozygous genotypes based on the *4 allele was 1.7% and 19%, respectively (Table 6 and Figure 5).

### 3.7. CYP2E1

The *1 allele frequency in the Iranian population was 94.1% [37]. The *4 allele, with a 5.6% frequency in the Iranian population, is the only one with a frequency of more than 1%. (Table 2, Figure 4). None of the Iranian ethnicities had the same frequency of the *4 allele as the European population (Table 3). At the same time, the Azeri was similar to the Middle Eastern population regarding the *4 allele frequency (Table 4) [46]. The Persian (8%) and the Azeri (3.5%) also had this allele’s highest and lowest frequency, respectively. The homozygote genotype was found only in Kurds and Lurs (1%), with no other ethnicities having this allele. In addition, the Persian (16%) and the Azeri (7%) had the highest and lowest frequency of heterozygote genotype, respectively (Table 5 and Figure 5). In the Iranian population, the mean frequency of CYP2E1 homozygous and heterozygous genotypes based on the *4 allele was 0.3% and 10.7%, respectively (Table 6 and Figure 5).

### 3.8. CYP3A4

In the Iranian population, the prevalence of the *1 allele was 99.7% [37]. The frequency of mutated alleles of the CYP3A4 enzyme was less than 0.5% among the Iranian population (Table 2).

### 3.9. CYP3A5

According to our previous research, the prevalence of the *1 allele in the Iranian population was 3.8% [37]. The *3 allele, with a 96.1% frequency in the Iranian population, was the only one with a frequency of more than 1%. (Table 2 and Figure 4). The frequency of the *3 allele among all Iranian ethnicities, except Persian, Turkmen, and Lur, was similar to the Caucasian race [55] while showing no similarity with the European (Table 3) or Middle Eastern populations (Table 4) [38,42]. The Persian (98.2%) and the Arab (94.8%) had this allele’s highest and lowest frequencies, respectively. Genotypic analysis of these ethnicities revealed that the Persian (99%) and Arab (88%) had the highest and lowest frequency of homozygote genotypes, respectively. In comparison, the Arab (11%) and Persian (0%) had the highest and lowest frequency of heterozygote genotype, respectively (Table 5 and Figure 5). In the Iranian population, the mean frequency of CYP3A5 homozygous and heterozygous genotypes based on the *3 allele was 93.9% and 5.8%, respectively (Table 6 and Figure 5).

The current and previous studies on eight major Iranian ethnicities found that Iranian ethnicities should be considered individually and personalized in personalized medicine. Moreover, the genetic metabolism of the Iranian population and its races may not be conducted based on previous research on other races or populations around the world. Further, remarkable variations are observed among Iranian ethnicities. Thus, studies on one or a few races do not accurately represent the Iranian population. Finally, we hope that more accurate studies on the genetic and phenotypic status of Iranian cities and people, precisely the physiologic and pathophysiologic information of the medical sciences on the metabolism and other processes in the human body, will be revealed, leading to personalized medicine based on the specific status of the individual to improve the efficacy and reduce the side effects and treatment costs.

## 4. Conclusions

A comparison of the most common SNPs in the Iranian population in eight major Iranian ethnicities revealed significant differences between these ethnicities. Although many previous studies on the polymorphisms of major CYP isoforms in the Iranian population were based on the population’s proximity to the Caucasians, the findings of this study indicate significant genetic differences among eight major Iranian ethnicities and with the Caucasian race, and European and Middle Eastern populations. In addition, there was variation in eight significant ethnicities of the Iranian population regarding the allele frequency and genotype of CYP450s. As a result, it can be concluded that Iranian ethnicities should be treated as a distinct population in future genotypic and phenotypic studies of CYP enzyme activity compared to other populations. Moreover, it should be noted that each ethnicity of the Iranian population has its own set of SNP frequency and genotypes. More ethnicity-specific data can help build a complete picture of genetic variations, resulting in more accurate dose adjustments in patients and, thus, faster progress in personalized medicine. Finally, it appears that we need to go a step further and analyze each individual to determine the exact genotype of a patient to select the best medical option and action for that individual. Besides studying the activity of CYP450 enzymes, it is crucial to study the phenotype of these enzymes and their genotype. Numerous studies have shown that, despite the normal genetic status of these enzymes in individuals, phenotypes and activities of these enzymes change due to environmental factors such as disease, diet, behavioral habits, and specific lifestyles. Hence, it appears inevitable that these enzymes’ genotype and phenotype correlation be studied simultaneously for greater confidence when analyzing CYP450 enzymes and higher accuracy in personalized medicine studies.

## Figures and Tables

**Figure 1 jpm-12-01848-f001:**
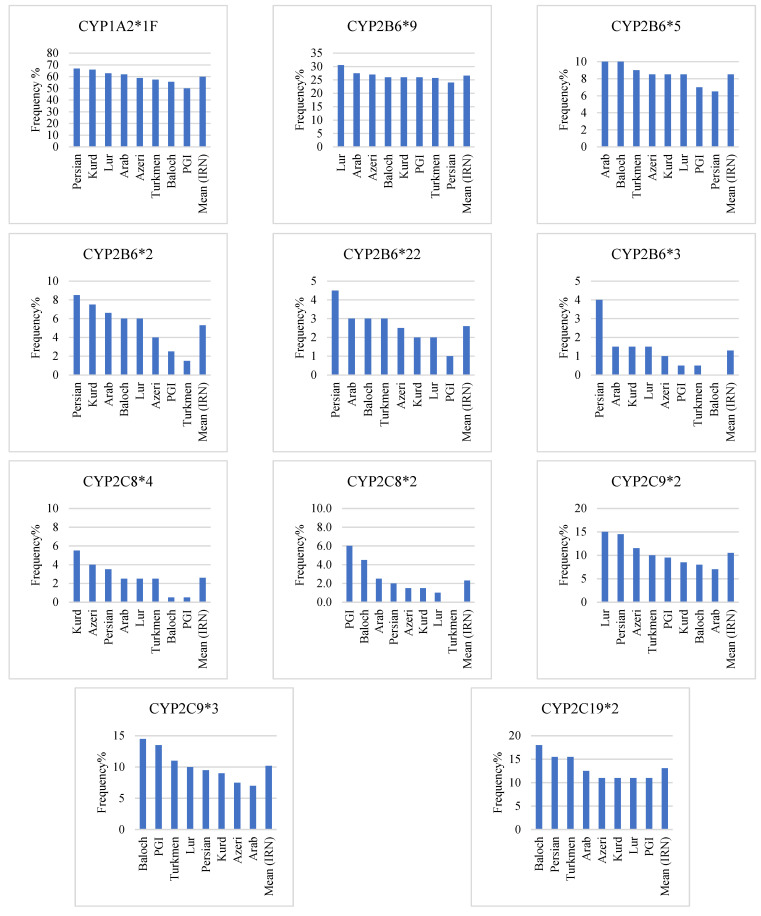
Comparison of most frequent SNPs in 8 major Iranian ethnicities and mean frequency in the Iranian population.

**Figure 2 jpm-12-01848-f002:**
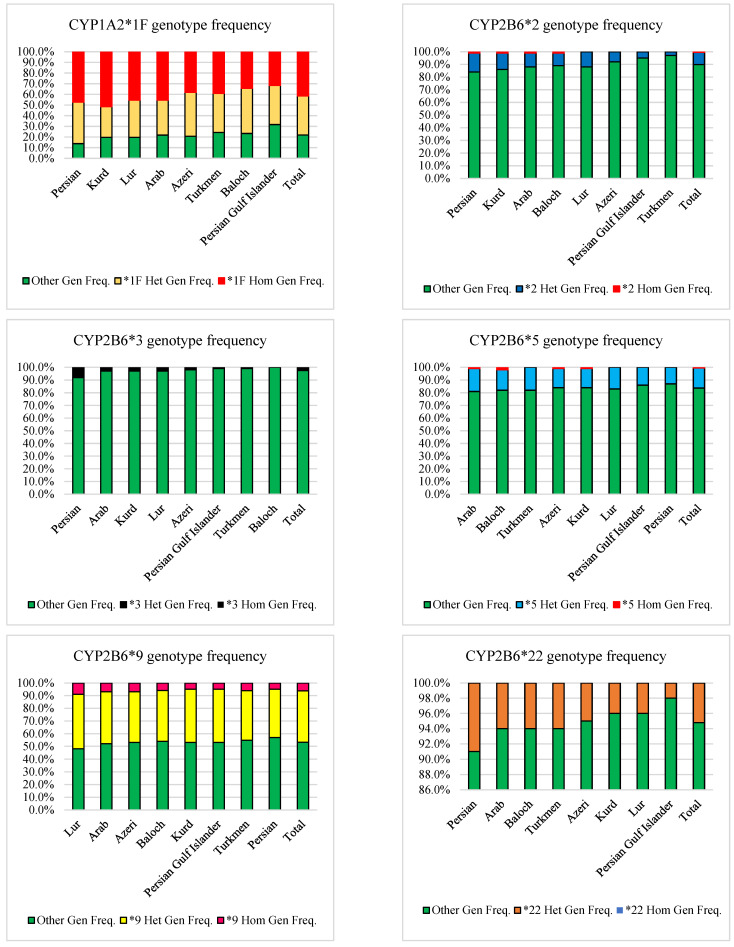
Comparison of CYP450s homozygous and heterozygous genotypes based on most frequent SNPs in major Iranian ethnicities and mean frequency in the Iranian population.

**Figure 3 jpm-12-01848-f003:**
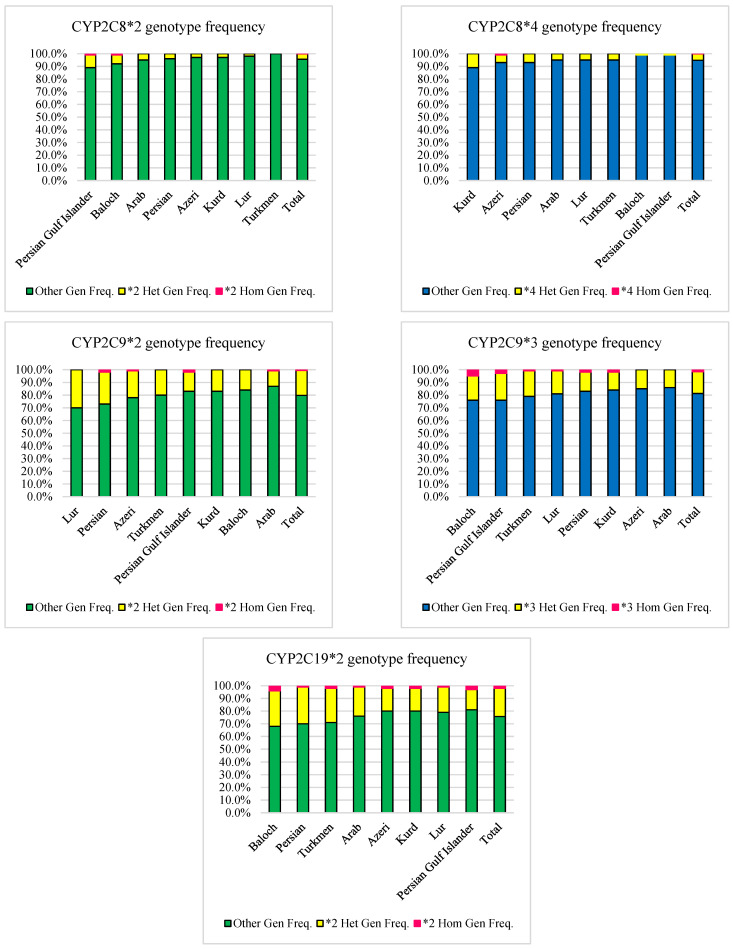
Comparison of CYP450s homozygous and heterozygous genotypes based on most frequent SNPs in major Iranian ethnicities and mean frequency in the Iranian population.

**Figure 4 jpm-12-01848-f004:**
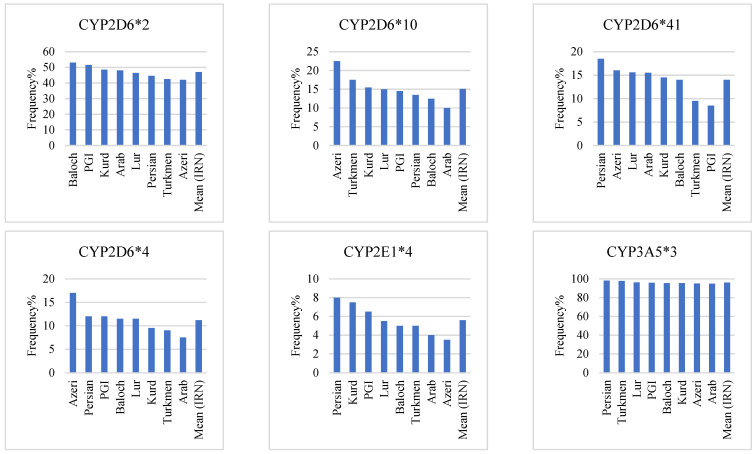
Comparison of most frequent SNPs in 8 major Iranian ethnicities and mean frequency in the Iranian population.

**Figure 5 jpm-12-01848-f005:**
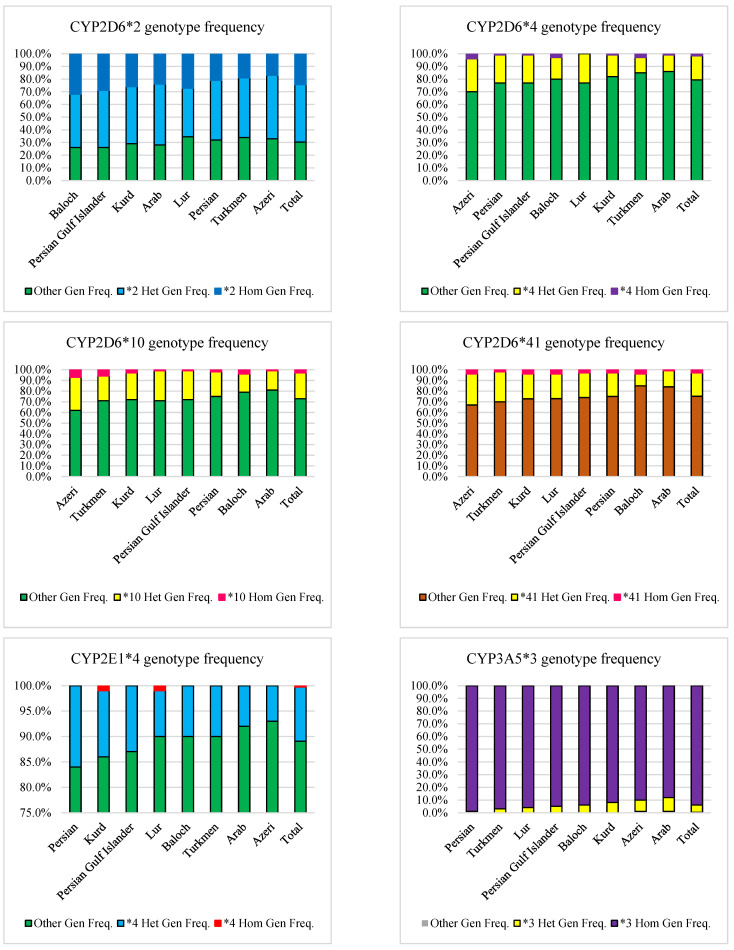
Comparison of CYP450s homozygous and heterozygous genotypes based on most frequent SNPs in major Iranian ethnicities and mean frequency in the Iranian population.

**Table 1 jpm-12-01848-t001:** Number of Genomes sequenced in 8 Iranian Ethnicities compared to the European population and Caucasian race.

Population	Ethnicities	Description	Genomes
IRN	Arab	Iranian (Based on (https://Iranome.ir, accessed on 10 August 2022))	100
IRN	Azeri		100
IRN	Baloch		100
IRN	Kurd		100
IRN	Lur		100
IRN	Persian		100
IRN	Persian Gulf Islander		100
IRN	Turkmen		100
European		Non-Finnish European (Based on (https://gnomad.broadinstitute.org, accessed on 10 August 2022))	~32,000
Middle Eastern		Middle eastern (Based on a literature review [38,39,40,41,42,43,44,45,46,47,48,49,50,51,52])	~50–200
-	Caucasian	Caucasian (Based on a literature review [6,7,10,14,37,53,54,55])	~100–250

**Table 2 jpm-12-01848-t002:** Prevalence of frequent alleles associated with nine CYP450 genes in major Iranian ethnicities.

	Ethnic								
Allele	Arab	Azeri	Baloch	Kurd	Lur	Persian	PGI *	Turkmen	Mean (IRN)
**CYP1A2*1F**	61.8	58.7	55.5	65.9	62.8	66.8	50	57.4	59.9
**CYP2B6*9**	27.6	27	26	26	30.5	24	26	25.7	26.6
**CYP2B6*5**	10	8.5	10	8.5	8.5	6.5	7	9	8.5
**CYP2B6*2**	6.6	4	6	7.5	6	8.5	2.5	1.5	5.3
**CYP2B6*22**	3	2.5	3	2	2	4.5	1	3	2.6
**CYP2B6*3**	1.5	1	0	1.5	1.5	4	0.5	0.5	1.3
**CYP2C8*4**	2.5	4	0.5	5.5	2.5	3.5	0.5	2.5	2.6
**CYP2C8*2**	2.5	1.5	4.5	1.5	1	2	6	0	2.3
**CYP2C9*2**	7	11.6	8	8.5	15	14.5	9.5	10	10.5
**CYP2C9*3**	7	7.5	14.5	9	10	9.5	13.5	11	10.2
**CYP2C19*2**	12.6	11	18	11	11	15.5	11	15.5	13.1
**CYP2D6*2**	48	42	53	48.5	46.4	44.5	51.5	42.5	47
**CYP2D6*10**	10	22.5	12.5	15.5	15	13.5	14.5	17.5	15.1
**CYP2D6*41**	15.5	16	14	14.5	15.6	18.5	8.6	9.5	14
**CYP2D6*4**	7.5	17	11.5	9.5	11.5	12	12	9	11.2
**CYP2E1*4**	4	3.5	5	7.5	5.5	8	6.5	5	5.6
**CYP3A4**	No allele frequency of more than 0.5%								
**CYP3A5*3**	94.8	95	95.5	95.4	96.3	98.2	95.9	97.7	96.1

* Persian Gulf Islander.

**Table 3 jpm-12-01848-t003:** The similarity of high-frequency alleles (SNPs) in major Iranian ethnicities with the Caucasian race and European population.

CYP450	Allele	IRN Ethnics Similar to Caucasian	Caucasian	IRN Ethnics Similar to European	European
CYP1A2	*1F (rs762551)	None	73.7% [55]	None	69.9%
CYP2B6	*9 (rs3745274)	Arab (27.6%)	28.6%	Persian (24%)	24%
	*5 (rs3211371)	Arab (10%)-Baloch (10%)	10.9%	None	11.8%
	*2 (rs8192709)	Baloch (6%)-Lur (6%)	5.3%	Arab (6.6%)-Baloch (6%)-Lur (6%)	5.6%
	*22 (rs34223104)	Arab (3%)-Baloch (3%)-Turkmen (3%)-Azeri (2.5%)-Kurd (2%)-Lur (2%)	2.4%	Kurd (2%)-Lur (2%)-Persian Gulf Islander (1%)	1.1%
	*3 (rs45482602)	Azeri (1%)-Persian Gulf Islander (0.5%)-Turkmen (0.5%)	<1%	Azeri (1%)-Persian Gulf Islander (0.5%)-Turkmen (0.5%)	<1%
CYP2C8	*4 (rs1058930)	None	7.5%	Kurd (5.5%)	5.4%
	*2 (rs11572103)	Lur (1%)	<1%	Lur (1%)	<1%
CYP2C9	*2 (rs1799853)	None	13.3%	Azeri (11.6%)	12.6%
	*3 (rs1057910)	None	5.6%	Arab (7%)-Azeri (7.5%)	6.8%
CYP2C19	*2 (rs4244285)	Arab (12.6%)	13.6%	Persian (15.5%)-Turkmen (15.5%)	14.6%
CYP2D6	*2 (rs16947, rs1135840)	All ethnics (42–53%)	32.8–52.5%	None	34.3%
	*10 (rs1065852, rs1135840)	None	19.6%	None	<1%
	*41 (rs28371725)	Turkmen (9.5%)-Persian Gulf Islander (8.6%)	9.6%	Turkmen (9.5%)	9.3%
	*4 (rs3892097)	None	18.2%	None	19.6%
CYP2E1	*4 (rs6413419)	-	NA	None	2.2%
CYP3A5	*3 (rs776746)	Baloch (95.5%)-Kurd (95.4%)-Persian Gulf Islander (95.9%)-Azeri (95%)-Arab (94.8%)	95.5%	None	93%

**Table 4 jpm-12-01848-t004:** The similarity of high-frequency alleles (SNPs) in major Iranian ethnicities with 4 Middle East countries.

CYP450	Allele	IRN Ethnics Similar to Middle Eastern	Turkish Pop.	Saudi Arabia Pop.	Egyptian Pop.	Emirati Pop.
CYP1A2	*1F (rs762551)	None	27%	-	68%	-
CYP2B6	*9 (rs3745274)	Lur (30.5)	11%	-	28.8%	30%
	*5 (rs3211371)	None	2%	-	3.8%	-
	*2 (rs8192709)	-	-	-	-	-
	*22 (rs34223104)	-	-	-	-	-
	*3 (rs45482602)	-	-	-	-	-
CYP2C8	*4 (rs1058930)	Arab (2.5%)-Lur (2.5%)-Turkmen (2.5%)	2.3%	-	-	-
	*2 (rs11572103)	-	-	-	-	-
CYP2C9	*2 (rs1799853)	Azeri (11.6%)-Turkmen (10%)	10.6%	13.3%	12%	11%
	*3 (rs1057910)	Arab (7%)-Azeri (7.5%)-Kurd (9%)-Lur (10%)-Persian (9.5%)-Turkmen (11%)	10%	2.3%	6%	7%
CYP2C19	*2 (rs4244285)	Baloch (18%)-Persian (15%)-Turkmen (15%)	18.3%	15%	3.8%	15%
CYP2D6	*2 (rs16947, rs1135840)	None	35%	10.4%	31.3%	12.2%
	*10 (rs1065852, rs1135840)	None	26%	3%	3.4%	3.3%
	*41 (rs28371725)	Arab (15.5%)-Azeri (16%)-Baloch (14%)-Kurd (14.5%)-Lur (15.6%)-Persian (18.5%)	15%	18.4%	15.1%	15.2%
	*4 (rs3892097)	Kurd (9.5%)-Turkmen (9%)	1%	3.5%	18.1%	9%
CYP2E1	*4 (rs6413419)	Azeri (3.5%)	-	-	2.8%	-
CYP3A5	*3 (rs776746)	None	3%	-	14%	-

**Table 5 jpm-12-01848-t005:** The frequency of CYP450s homozygous and heterozygous genotypes based on the most frequent SNPs in major Iranian ethnicities.

	Ethnic	1A2*1F	2B6*2	2B6*3	2B6*5	2B6*9	2B6*22	2C8*2	2C8*4	2C9*2	2C9*3	2C19*2	2D6*2	2D6*4	2D6*10	2D6*41	2E1*4	3A5*3
**HOMOZYGOUS**	**Arab**	45.3%	1.0%	0.0%	1.0%	7.0%	0.0%	0.0%	0.0%	1.0%	0.0%	1.0%	24.0%	1.0%	1.0%	1.0%	0.0%	88.0%
	**Azeri**	38.1%	0.0%	0.0%	1.0%	7.0%	0.0%	0.0%	1.0%	1.0%	0.0%	2.0%	17.0%	4.0%	7.0%	4.0%	0.0%	90.0%
	**Baloch**	34.3%	1.0%	0.0%	2.0%	6.0%	0.0%	1.0%	0.0%	0.0%	5.0%	4.0%	32.0%	3.0%	4.0%	4.0%	0.0%	93.9%
	**Kurd**	51.5%	1.0%	0.0%	1.0%	5.0%	0.0%	0.0%	0.0%	0.0%	2.0%	2.0%	26.0%	1.0%	3.0%	4.0%	1.0%	92.0%
	**Lur**	45.3%	0.0%	0.0%	0.0%	9.0%	0.0%	0.0%	0.0%	0.0%	1.0%	1.0%	27.2%	0.0%	1.0%	4.0%	1.0%	96.0%
	**Persian**	47.3%	1.0%	0.0%	0.0%	5.0%	0.0%	0.0%	0.0%	2.0%	2.0%	1.0%	21.0%	1.0%	2.0%	3.0%	0.0%	99.0%
	**PGI ***	31.6%	0.0%	0.0%	0.0%	5.0%	0.0%	1.0%	0.0%	2.0%	3.0%	3.0%	29.0%	1.0%	1.0%	3.0%	0.0%	95.0%
	**Turkmen**	39.0%	0.0%	0.0%	0.0%	6.1%	0.0%	0.0%	0.0%	0.0%	1.0%	2.0%	19.0%	3.0%	6.0%	2.0%	0.0%	97.0%
**HETEROZYGOUS**	**Arab**	32.9%	11.0%	3.0%	18.0%	41.0%	6.0%	5.0%	5.0%	12.0%	14.0%	23.0%	48.0%	13.0%	18.0%	15.0%	8.0%	11.0%
	**Azeri**	41.2%	8.0%	2.0%	15.0%	40.0%	6.0%	3.0%	6.0%	21.0%	15.0%	18.0%	50.0%	26.0%	31.0%	29.0%	7.0%	9.0%
	**Baloch**	42.4%	10.0%	0.0%	16.0%	40.0%	6.0%	7.0%	1.0%	16.0%	19.0%	28.0%	42.0%	17.0%	17.0%	11.0%	10.0%	6.1%
	**Kurd**	28.8%	13.0%	3.0%	15.0%	42.0%	4.0%	3.0%	11.0%	17.0%	14.0%	18.0%	45.0%	17.0%	25.0%	23.2%	13.0%	8.0%
	**Lur**	35.0%	12.0%	3.0%	17.0%	43.0%	4.0%	2.0%	5.0%	30.0%	18.0%	20.0%	38.3%	23.0%	28.0%	23.0%	9.0%	96.0%
	**Persian**	38.9%	15.0%	8.0%	13.0%	38.0%	9.0%	4.0%	7.0%	25.0%	15.0%	29.0%	47.0%	22.0%	23.0%	22.0%	16.0%	99.0%
	**PGI ***	36.7%	5.0%	1.0%	14.0%	42.0%	2.0%	10.0%	1.0%	15.0%	21.0%	16.0%	45.0%	22.0%	27.0%	23.0%	13.0%	95.0%
	**Turkmen**	36.7%	3.0%	1.0%	18.0%	39.1%	6.0%	0.0%	5.0%	20.0%	20.0%	27.0%	47.0%	12.1%	23.0%	28.0%	10.0%	97.0%

***** Persian Gulf Islander.

**Table 6 jpm-12-01848-t006:** The frequency of CYP450s homozygous and heterozygous genotypes based on the most frequent SNPs in the Iranian population.

	Genotype	
Allele	Homozygous	Heterozygous
**CYP1A2*1F**	41.5%	36.6%
**CYP2B6*2**	0.5%	9.6%
**CYP2B6*3**	0.0%	2.6%
**CYP2B6*5**	0.6%	15.7%
**CYP2B6*9**	6.2%	40.6%
**CYP2B6*22**	0.0%	5.2%
**CYP2C8*2**	0.2%	4.2%
**CYP2C8*4**	0.1%	5.1%
**CYP2C9*2**	0.7%	19.5%
**CYP2C9*3**	1.7%	17.0%
**CYP2C19*2**	2.0%	22.3%
**CYP2D6*2**	24.4%	45.3%
**CYP2D6*4**	1.7%	19.0%
**CYP2D6*10**	3.1%	24.0%
**CYP2D6*41**	3.1%	21.7%
**CYP2E1*4**	0.3%	10.7%
**CYP3A5*3**	93.9%	5.8%

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
