# Peer review of "Variations in the Frequencies of Polymorphisms in the CYP450s Genes in Eight Major Ethnicities of Iran: A Review of the Human Data"

_jpm, 2022, doi:10.3390/jpm12111848_

Round 1

Reviewer 1 Report

This is a nice work which presented a review of germline SNPs in the CYP450s gene in major ethnicities from the Iranian population, and then compared the allele frequency in the population database such as GnomAD. I have few comments and suggestion to improve the review article

·      In the method the author described common alleles with MAF > 1%. Is the genotype for common allele considered as Homozygous alternate and Heterozygous?

·      The table 2 and Figure 1a is redundant and displaying similar results. I would suggest keeping the table which is more informative

·      Could author show scatter plot rather than table3 for allele frequency of variants in the Iranian population with the Caucasian and European? If the allele frequency between the study population is similar to the rest of race/population database, then the significant correlation will be achieved with corr value > 0.9?

·      In the Figure 2a, the author showed the comparison of CYP450s homozygous and heterozygous genotypes in the Iranian ethnicities. The author claimed that Baloch and Kurds have the highest and lowest allele frequency for CYP1A2*1F. Could author support the result with statistical test to show significant increase or decrease?

·      Is any of the CYP450 genes have an association with disease/ cancer formation and progression? If yes, what is the mentioned in the context of the Iranian population? Could you please describe in the discussion section?

Author Response

  • In the method the author described common alleles with MAF > 1%. Is the genotype for common allele considered as Homozygous alternate and Heterozygous?

Answer: We have gathered the data of Iranian population from Iranome database and in this website the genotype of Iranian population based on these frequent alleles has been reported. So, yes. The genotype for common alleles considered as Homozygous and Heterozygous.

  • The table 2 and Figure 1a is redundant and displaying similar results. I would suggest keeping the table which is more informative

Answer: We demonstrated the comparison of most frequent SNPs in 8 major Iranian ethnicities and mean frequency in the Iranian population in figure 1a in order to show the table 2 data more illustrative. But if you think its better to be deleted, we will do it.

  • Could author show scatter plot rather than table3 for allele frequency of variants in the Iranian population with the Caucasian and European? If the allele frequency between the study population is similar to the rest of race/population database, then the significant correlation will be achieved with corr value > 0.9?

Answer: Actually because the similarity between Iranian ethnicities and European population and Caucasian race was low, we demonstrated the data as table 3 to show the differences between these populations better. Most of Iranian ethnics are not similar to European population and Caucasian race according to allele frequencies.

  • In the Figure 2a, the author showed the comparison of CYP450s homozygous and heterozygous genotypes in the Iranian ethnicities. The author claimed that Baloch and Kurds have the highest and lowest allele frequency for CYP1A2*1F. Could author support the result with statistical test to show significant increase or decrease?

Answer: We have demonstrated the allele frequency of CYP450s in table 2 and as you can see the highest and lowest CYP1A2*1F allele frequency are among Persian and Persian Gulf Islander ethnics in the Iranian population. In figure 2a we compared the CYP450s homozygous and heterozygous genotypes based on most frequent SNPs in major Iranian ethnicities and for example the Kurd and Persian Gulf Islander ethnics have the highest and lowest homozygous (CYP1A2*1F/*1F) frequency.

  • Is any of the CYP450 genes have an association with disease/ cancer formation and progression? If yes, what is the mentioned in the context of the Iranian population? Could you please describe in the discussion section?

Answer: As you know there are many disease/ cancer formation and progression that associated with the CYP450 genotype/phenotype state. But, because in this manuscript we only reported the allele frequency and genotype frequency of Iranian 8 major ethnicities and compared them with each other and other populations in order to show the differences of this ethnics with each other and other populations, we didn't discuss about clinical outcomes of these results. the aim of this study is to demonstrate that as Personalized medicine sciences tell us, we should consider the differences even in a country population. Maybe in the future we publish a manuscript and fully evaluate the impact of these genotype in the Iranian population clinical outcomes.

Reviewer 2 Report

The work “Variations in the frequencies of polymorphisms in the CYP450s genes in eight major ethnicities of Iran: A review of the human data” by Navid Neyshaburinezhad et al. is very interesting. The research object focused on the CYP450s genes of Iran makes the research more typical. As a review article, the related research works were comprehensively and exhaustively discussed, further, this review article was well organized and presented. There are some minor suggestions for the authors.

1. line 12 of page 2: “genotype and phe-” should be “Genotype and phe”. Capitalize the first letter.

2. As parts of a series of studies or important researches by the authors, the related articles https://doi.org/10.1016/j.phrs.2016.07.002 ,  https://doi.org/10.1002/bdd.2107 and https://doi.org/10.1007/s40291-013-0028-5  should also be cited.

3. A reference or web site is needed for “Based on literature review” in Table.1, and it will make the material and methods more rigorous.

4. The first sentence of the “Results and Discussion” section need some references at the proper position because the author claimed that “Several studies have reported the frequency of different CYP alleles in …”.

5. For the reader's convenience, the figure.1b should located before figure.2a?

Author Response

line 12 of page 2: “genotype and phe-” should be “Genotype and phe”. Capitalize the first letter.

Answer: It has been corrected and also, the manuscript has undergone English revision.

As parts of a series of studies or important researches by the authors, the related articles https://doi.org/10.1016/j.phrs.2016.07.002 ,https://doi.org/10.1002/bdd.2107 and https://doi.org/10.1007/s40291-013-0028-5  should also be cited.

Answer: We have cited above articles in the manuscript.

A reference or web site is needed for “Based on literature review” in Table.1, and it will make the material and methods more rigorous.

Answer: References have been added to the mentioned section.

The first sentence of the “Results and Discussion” section need some references at the proper position because the author claimed that “Several studies have reported the frequency of different CYP alleles in …”.

Answer: References have been added to the mentioned section.

For the reader's convenience, the figure.1b should located before figure.2a?

Answer: It has been corrected as mentioned above.